# MicroRNA-222 Regulates Melanoma Plasticity

**DOI:** 10.3390/jcm9082573

**Published:** 2020-08-08

**Authors:** Maria Chiara Lionetti, Filippo Cola, Oleksandr Chepizhko, Maria Rita Fumagalli, Francesc Font-Clos, Roberto Ravasio, Saverio Minucci, Paola Canzano, Marina Camera, Guido Tiana, Stefano Zapperi, Caterina A. M. La Porta

**Affiliations:** 1Center for Complexity and Biosystems, Department of Environmental Science and Policy, University of Milan, via Celoria 26, 20133 Milano, Italy; maria.lionetti@unimi.it (M.C.L.); maria.fumagalli@unimi.it (M.R.F.); 2Center for Complexity and Biosystems, Department of Physics, University of Milan, via Celoria 16, 20133 Milano, Italy; filippo.cola@unimi.it (F.C.); franscesc.font@unimi.it (F.F.-C.); guido.tiana@unimi.it (G.T.); stefano.zapperi@unimi.it (S.Z.); 3Institut für Theoretische Physik, Leopold-Franzens-Universität Innsbruck, Technikerstrasse 21a, A-6020 Innsbruck, Austria; oleksandr.chepizko@gmail.com; 4CNR-Consiglio Nazionale delle Ricerche, Biophysics institute, via De Marini 6, 16149 Genova, Italy; 5Department of Experimental Oncology, European Institute of Oncology, Via Adamello 16, 20139 Milano, Italy; roberto.ravasio@unimi.it (R.R.); saverio.minucci@unimi.it (S.M.); 6Centro Cardiologico Monzino I.R.C.C.S., Via Carlo Parea 4, 20138 Milano, Italy; paola.canzano@cardiologicomonzino.it (P.C.); marina.camera@unimi.it (M.C.); 7Department of Pharmacological and Biomolecular Sciences, University of Milan, Via Balzaretti 9/11/13, 20133 Milano, Italy; 8CNR-Consiglio Nazionale delle Ricerche, Istituto di Chimica della Materia Condensata e di Tecnologie per l’Energia, Via R. Cozzi 53, 20125 Milano, Italy; 9Innovation for Well-Being and Environment (CR-I-WE), University of Milan, via Celoria 26, 20133 Milano, Italy

**Keywords:** melanoma, tumor plasticity, hsa-mir-222, reaction diffusion, kinetic equations, cancer stem cells, phenotypic switching

## Abstract

Melanoma is one of the most aggressive and highly resistant tumors. Cell plasticity in melanoma is one of the main culprits behind its metastatic capabilities. The detailed molecular mechanisms controlling melanoma plasticity are still not completely understood. Here we combine mathematical models of phenotypic switching with experiments on IgR39 human melanoma cells to identify possible key targets to impair phenotypic switching. Our mathematical model shows that a cancer stem cell subpopulation within the tumor prevents phenotypic switching of the other cancer cells. Experiments reveal that hsa-mir-222 is a key factor enabling this process. Our results shed new light on melanoma plasticity, providing a potential target and guidance for therapeutic studies.

## 1. Introduction

Plasticity of tumor cells is not only a key ingredient of tumor growth but also a critical step for developing a successful anti-tumor therapy. Recently, our group showed that a network of miRNAs modulates phenotypic switching in human melanoma through Wnt signalling and epithelial–mesenchymal transition (EMT) in human melanoma cells, controlling the total number of cancer stem cells (CSCs) in the tumor [1]. Among the identified miRNAs, hsa-mir-222-5p plays a critical role [1]. The ability of CSCs to keep their number constant resembles a critical biological feature of stem cells (SCs). Since SCs live in a confined space (the niche), they need to ensure that their number is kept constant [2]. If for some reason a drastic depletion of SCs occurs, other more differentiated cells can revert to the SC state, replenishing the population. During this process, the CSCs subpopulation overshoots to a concentration higher than the original one and then returns to the initial level. The classical cell hierarchy within the tissue considers SCs at the top of a lineage and their differentiation as an irreversible process. In recent years, this paradigm has been challenged by numerous studies demonstrating that progenitors and differentiated cells in different tissues, from intestine and gastric cells to skin and lung, display considerable plasticity under various conditions [3,4,5,6,7,8]. Furthermore, it has been shown that niche-derived signals contribute to this plasticity [9,10,11,12]. Recent work indeed showed that microenvironmental signals [13] and the density of the extracellular matrix [14] modulate the spatial patterning of CSCs. One of the important consequences of tumor plasticity lies in its impact on drug treatment [15]. Interestingly, a recent paper showed that after continuous drug treatment, slow-cycling melanoma cells enriched within the tumor show a SC-like phenotype [16].

In this paper, we investigated the regulatory mechanisms controlling phenotypic switching in human melanoma IgR39 cells, by combining mathematical models and experiments. We first developed a mathematical model to illustrate how phenotypic switching of cancer cells (CCs) into CSCs can be controlled by the release of switch-inhibiting factors from CSCs. Previous work from our group showed that CSCs in melanoma can be identified using the CXCR6 marker [1,17]. We have also already demonstrated that a stochastic mechanism displaying an environment-independent switching rate would not account for the observed overshoot and we proposed a model which introduces a switching rate that depends on an activating miRNA, whose dynamics are controlled by the exponential of the fraction of CSCs, in agreement with experiments [1]. Even if this model could reproduce experimental observations, it was difficult to translate it into a specific molecular mechanism. In the literature, there are other models that use rate equations to describe the growth of the tumor mass, including phenotypic switching [18,19,20,21,22]. For example, in [19] a key ingredient to describe the de-differentiation of epithelial cells to the more aggressive mesenchymal phenotype is a time delay of some days, resulting in delayed differential equations.

Here we describe an alternative model to control phenotypic switching that can easily be interpreted in terms of molecular interactions in agreement with previous experimental data. Unlike other models, we suggest that phenotypic switching is a spontaneous tendency of all cancer cells, and that, under standard conditions of tumor growth, switching is inhibited by a specific molecular species produced by CSCs. We then investigated further the involvement of hsa-mir222-5p as one of the key factors responsible for the switch of melanoma cells. We, therefore, knocked down hsa-mir222-5p, showing that IgR39 cells loose the capability to revert the expression of CSC’s markers and acquire a less aggressive phenotype, as shown by measuring the migration capabilities of the cells in 2D wound healing assays and in 3D spheroids invasion assays.

Altogether, our results shed a new light on tumor plasticity and provide guidance for further targeted studies aimed at impairing it.

## 2. Results

### 2.1. Overshoot in the Mean-Field Model

We developed a mean-field model in which CSCs are capable of unlimited proliferation [1,23], while CCs can only divide into two CCs for a limited number *g* of generations, after which they become senescent and die. Details and equations of the model are reported in the Materials and Methods section and in Appendix B.

The model was used to generate the dynamical behavior of the number *S* of CSCs, of the number $N_{j}$ of CCs with various age *j*, of the total number $N=\sum_{i=0}^{g}N_{i}$ of CCs and of the number *m* of inhibitor molecules. A first experimental observation that the model should reproduce is that the total number of cells S+N must be an increasing and unbounded function of time. A necessary and sufficient condition for this is that
(1)$$\epsilon \equiv k_{2S}-k_{2N}\geq 0.$$

Once the condition is satisfied, we can compare with experiments the fraction $c=\frac{S}{S+N}$ of CSCs over the total number of cells. This quantity has been observed [1] to experience an overshoot ten days after sorting in a population initially devoid of CSCs. Furthermore, the amount of CSCs in the overshoot peak decreases with the fraction c(0) of CSCs remaining in the initial population, as shown in Figure 1b (red bars).

The set of parameters associated with an overshoot which are most similar to the experimental data, is given in Table A1, corresponding to dynamics of c(t) and to a dependence on c(0) described by the continuous lines in Figure 1b. The initial conditions for these simulations imply an initial number of total cells of $10^{6}$[1], with an initial fraction of sorted CSCs ranging from 0.001 (0.1%) to 0.01 (1%). CCs are assumed to be distributed among the *g* generations reflecting the steady-state distribution among generations associated with our model. We assume that is m(0)=0 during the sorting procedure when the inhibiting molecule is washed out (although a small amount of m(0) does not change qualitatively the results). The overshoot results to be sizable if the sorting procedure leaves up to 1% of CSCs in the initial set of cells (see Figure 1c), in agreement with experiments [1]. Consequently, there is at least a set of parameters for which the model reproduces qualitatively the experimental data. Being a thorough exploration of the whole parameter space of the model computationally unfeasible, we started from those of Table A1 and studied the dynamics of the system varying one parameter at a time.

The value of the maximum $c_{max}$ concentration of CSCs during the overshoot and of its stationary value $c^{*}$ are displayed in Figure A1, as function of some of the parameters of the system. The size of the overshoot increases smoothly but considerably as a function of the switching rate σ and of the rate difference ϵ, and decreases as a function of the aging rate $k_{d}$ of CCs. The dependence on the other parameters is much weaker. These results suggest that the overshoot is overall a robust property of the model. The strong dependence on the switching rate σ indicates that phenotypic switching is an important ingredient to reproduce the experimental observations. Two important parameters seem to be $k_{d}$ and *g*, that control together the total amount of CCs. For large values of $k_{d}$ or *g*, the overall number of CC is large and the overshoot effect disappears. A simple qualitative explanation for this is that if we increase the values of $k_{d}$ or *g*, then the total number of CC cells increases exponentially, while the number of CSCs increases only as there are more CC cells able to switch. Therefore we expect the increase in those values to reduce the amount of the overshoot.

### 2.2. Inhibition of Phenotypic Switching

Since the phenotypic switching is inhibited by a molecular species $m_{0}$ produced by CSCs, it should be possible to inhibit phenotypic switching after sorting out CSCs by growing the cells in a medium containing the inhibiting molecule. To understand this idea from a quantitative point of view, we performed simulations according to Equation (Equation 3) and compared the total number of cells at fixed time *T* when the sorted cells are treated at time t=0 with a solution containing $m_{0}$ inhibiting molecules, with the total number of cells in the untreated case (i.e., $m_{0}=0$). We defined the quantity
(2)$$R_{T,m_{0}}=\frac{{\left[S\left(T\right)+N\left(T\right)\right]}_{m\left(0\right)=m_{0}}}{{\left[S\left(T\right)+N\left(T\right)\right]}_{m(0)=0}}$$
to quantify the effect of the inhibiting molecule at time *T*. It should be noted that the amount of inhibiting molecule varies with time from its initial value, because, besides that provided at time zero, it is produced by CSCs and undergoes degradation.

In Figure A2a, we show the variation of $R_{T,m_{0}}$ as a function of the time *T* after sorting. The figure shows that a noticeable effect can be seen already a few days after sorting, although a strong effect could be seen after 20 days, when the number of cancer cells is reduced by about 50% for the highest initial concentrations of the inhibiting molecule. In Figure A2b, we also display the value of $R_{20days,m_{0}}$ as a function of $m_{0}$, for different choices of $c_{0}$, evaluating the number of cells after 20 days. Independently of the remaining concentration $c_{0}$ of CSCs, there is a threshold of ∼10${}^{5}$ molecules, beyond which one can observe an inhibitory effect. Assuming that the plate used for the experiment has a volume of ∼10 mL, this threshold corresponds to a concentration of $10^{-17}$ mol, meaning that even a tiny amount of molecule has a detectable effect on the (exponential) growth of the cells. The effect of the inhibitor saturates to a $c_{0}$-dependent value when its copy number reaches ∼10${}^{12}$, corresponding to ∼1 nmol.

### 2.3. Simulations of Tumor Growth in Two Dimensions

The rate Equations (Equation 3) average in space the concentration of cells and of the inhibitor factor. To investigate whether specific spatial pattern appear during tumor growth, we studied an extension of the model in which the spatial degrees of freedom appear explicitly.

We assume that cells sit on the vertexes of a square lattice, whose elementary length is then *ℓ*∼10μm and whose linear size is typically $10^{3}\ell$∼1 cm. Each site can host at most one cell, either normal or CSC. It undergoes the same processes (symmetric or asymmetric division, phenotypic switching, cellular aging and death, production of the switching-inhibitor molecule) as described in Figure 1a for the rate-equation model. Also the numerical values assigned to the rates of the different processes are the same. The cellular dynamics is simulated with a Gillespie algorithm [24]. Upon cell division, one of the new cells remains in the same site, while the other occupies a neighboring site. If all the neighboring sites are occupied, all cells in a random direction are shifted to make room for the new one. Neighbors are defined along the horizontal, vertical and diagonal directions, according to the Moore scheme, which minimizes artifacts due to the square lattice [25]. A further parameter which was not defined in the mean-field model is the diffusion coefficient of the inhibiting factor. Assuming that it is a nm-sized miRNA complex, Stokes’ law would suggest a diffusion coefficient *D*∼100μm${}^{2}$/s, which would decrease to *D*∼0.1μm${}^{2}$/s in case it is encapsulated into a vesicle.

Simulating the diffusion of the inhibiting factor within the Gillespie algorithm is computationally unfeasible, because each movement of every single molecule takes place on a time scale which is much smaller than that associated with the cellular processes, and thus essentially all the computational time would be used to simulate microscopic diffusion processes. Consequently, diffusion is simulated in parallel to the Gillespie algorithm by solving a discretized version of the diffusion equations as discussed in the Materials and Methods section. Simulation of the dynamics of this model starting from a set of cells purified from CSCs display an overshoot in CSCs concentration, similar to what was observed in the rate equations and in the experiments.

We then focused our analysis on the growth of a tumoral mass starting from a single CSC, as displayed in Figure 2a. The overall growth of the number of CCs and CSCs becomes exponential after ∼5 days with a rate that depends mildly on the diffusion coefficient of the molecule (from 0.4 days${}^{-1}$ at low *D* to 0.5 days${}^{-1}$ at large *D*) and is indistinguishable from a model without phenotipic switching. The fraction S/(S+N) of CSCs decreases to a stationary value of few percent with a weak dependence on *D*.

Unlike the cellular growth rate and the fraction of CSCs, the spatial organization of CCs and CSCs depends markedly on the properties of the diffusing molecule. Figure 2a shows that for higher values of the diffusion constant, the clusters of CCs are more dense. The most notable feature of is the accumulation of CSCs at the border of the tumor. This effect increases with the diffusion coefficient *D* of the molecule and is absent if no switching applies. This effect can be better quantified from the radial distribution of the fraction of CSCs displayed in Figure 2b. When *D* is large, the fraction of CSCs at the border of the tumor displays a higher peak than at smaller values of *D*, while no enrichment at all is observed in the absence of switching.

From the snapshots displayed in Figure 2a it is apparent that for small and intermediate values of *D*, CSCs tend to cluster together, especially at the border. To get a more quantitative insight in this, we performed a clustering analysis of CSC based on their spatial density [26]. In Figure 2c we show the multiplicity of clusters found and the distribution of their density. Without switching, the system displays a small number of high-density clusters. A similar behavior is observed with switching at low diffusion constant. Increasing *D*, we observe many low-density clusters, suggesting that CSCs decrease their tendency to partition in clusters.

Summing up, clustering seems to appear as a consequence of CSCs duplication, while switching, which is more effective at large *D*, tends to make the distribution of CSCs more uniform. This is not unexpected, since cell duplication is a local phenomenon, while switching is random in space, and can only be modulated in space by the diffusing factor. The high diffusivity of the factor makes, however, its distribution to be quite uniform, suppressing most spatial patterns.

### 2.4. Impact of hsa-mir222-5p on Phenotopic Switching

According to the model when CSCs are sorted out by flow cytometry, CCs are not exposed anymore to the switch inhibiting factor and therefore start to massively switch in agreement with our previous observations [1]. In order to better understand the mechanism underlying the model hypothesis, we collected the conditioned media (CM) of IgR39 cells and analyzed the level of expression of miRNAs. As shown in Figure A3, there is a good correlation between the expression of miRNAs inside the cell and the amount of miRNAs found in the conditioned media. Since hsa-mir222-5p was strongly involved in phenotypic switching [1], we knocked it down in IgR39 cells by pEGFP–Sponge-mir222-5p (sh-mir222, see Materials and Methods). As shown in Figure A4, the level of expression of hsa-mir222-5p was significantly lower with respect to cells infected with a scramble miRNA lentivirus. Moreover under the same conditions, a phenotypic marker of CSCs, CXCR6, was expressed at very low level, see Figure A5. Finally, we checked if the knock down of hsa-mir222-5p had an impact on other miRNAs. As shown in Figure A6, no significant changes in expression were detected in other miRNAs (see Materials and Methods for more details).

### 2.5. Impact of hsa-mir222-5p on Cell Migration

Epithelial–mesenchymal transition (EMT) is a critical feature that helps tumor cell spreading by acquiring a higher motility [27]. Herein, we checked if the lack of expression of hsa-mir222-5p had an impact on cell migration using two different approaches: (a) in a 2D wound healing assay and (b) in 3D spheroids quantifying the invasion of the cells grown as spheroids into a collagen network [28]. In both 2D or 3D models we found a decrease in motility of the cells without hsa-mir222-5p, confirming the role of this miRNA in mesenchymal phenotype [1]. In particular, Figure 3a shows two snapshots of the local velocities and vorticities obtained by the PIV method where arrows represent velocities and color indicates vorticity. The snapshots show that upon hsa-mir222-5p knockdown, velocities tend to decrease. This is confirmed by computing the distributions of velocities that we report in Figure 3b.

Next, we carried out experiments growing the cells in spheroids embedded in collagen networks according to the protocol discussed in the Materials and Methods [28]. Time lapse observations are summarized in Figure 3c for scramble and sh-mir222 knockdown cells (see also Appendix A and Appendix D). We can see that while scramble cells spread into the matrix, sh-mir222 knockdown cells remain confined into the spheroid. While cells do not spread, we observe motion of cells within the spheroid (see Appendix A). This implies that while motility is still present, invasion capabilities are strongly impaired. We quantified these observations by measuring the typical spread of the spheroid as ${\left(\langle {\left(R-R_{c}\right)}^{2}\rangle \right)}^{1/2}$, where $R_{c}$ is the center of mass of the spheroid. Figure 3d summarizes the results.

## 3. Discussion

Melanoma is a highly resistant tumor with poor patient survival due to its intrinsically high metastatic capability [27]. The appearance of resistance after chemotherapy or the presence of intrinsic resistance leads to great difficulties in devising an effective and durable therapy. The ability of tumor cells to change their status using epigenetic mechanisms independent of the environment, such as the tumor niche, has been shown to play a critical role in the acquisition of a resistant phenotype in response to specific drugs. In light of these findings, the ability of tumor cells to change their phenotype, becoming resistant, can happen at the level of the tumor cells and/or at the level of the tumor niche. Interfering with this mechanism appears to be a promising and crucial issue to fight metastasis [27].

Our group recently demonstrated that human melanoma cells can change their phenotype, expressing EMT markers dynamically and activating Wnt-β-catenin pathway, thanks to a complex network of miRNAs which includes hsa-mir-222-5p [1]. The most important consequence of those findings is that tumors can control the number of CSCs in the bulk. In fact, these factors drive the switch of CCs into CSCs under specific environmental conditions [1,27]. It is clear that one interesting strategy could be to interfere with this process, finding a way to stop phenotypic switching to CSCs. This was the main goal of the present study. To better understand the regulation of the number of CSCs in the tumor cell population, we resorted to mathematical models. We first considered a mean-field model subdividing cancer cells into different compartments corresponding to CSCs and CCs at different generations. While CSCs can self-renew or give rise to CCs and CCs divide only for a finite number of generations, we considered the possibility for CCs to revert into the CSC state. This possibility is modulated by an inhibiting factor released by CSCs. With these assumptions, the model is able to faithfully reproduce the phenotypic switching observed experimentally [1].

We have then investigated a putative candidate for the inhibiting factor: hsa-mir-222-5p. This miRNA is strongly involved in phenotypic switching, being highly expressed before the overshoot of CSCs [1]. Here, we found that this factor is also present in the conditioned media (CM), in agreement with recent observations in melanoma [29]. Knockdown of hsa-mir-222-5p has important consequences for tumor aggressiveness: (i) the cells no longer express CSC markers and (ii) their migration capability is impaired.

We can reconcile our model with our observations considering a feedback loop driven by hsa-mir-222 (see Figure 4). In the steady-state, hsa-mir-222 is expressed by the cells and released outside, acting on paracrine cells. We reported in Ref. [1] that in unsorted melanoma cells, the level of expression of LEF1 shows a two fold increase upon silencing of hsa-mir-222 (Figure 4). LEF1 is a key downstream target of the Wnt signaling pathway. The canonical Wnt signaling is activated when the Wnt ligand binds the Frizzled and LRP5/6 receptors. This leads to the stabilization of β-catenin which is transferred from the cytoplasm to the nucleus where it activates LEF1 (see Figure 4). It is known that MITF, which is directly induced by LEF1, acts as an activator of hsa-mir-222 [30]. Hence, activation of the Wnt pathway leads to increased hsa-mir-222 expression. On the other hand, it has been reported that hsa-mir-222 can inhibit the translation of Frizzled [31,32], thus suppressing the activity of the pathway (Figure 4). Accordingly, as already shown in [1] and reported in Figure 4, CCs three days after sorting, at the onset of phenotypic switching, display a level of LEF1 similar to the one in unsorted cells. When hsa-mir-222 was silenced, LEF1 decreased slightly. Taken together, these results suggest that the switch inhibitor is hsa-mir-222 itself within the feedback loop illustrated in Figure 4.

In addition to mean-field reaction equations, we have also studied a two-dimensional variant of the model where we considered the interplay between tumor growth, phenotypic switching and the diffusion of the switch inhibitor. Simulations of the model showed when phenotypic switching is present, CSCs cluster at the boundary of the tumor mass in a diffusion-dependent manner. This result is interesting because it provides a natural mechanism that could help the dissemination of CSCs into secondary tumors [27]. Notice that in a previous spatial model of cancer growth with CSCs, the localization of CSCs at the border was imposed by reducing the motility of the other cells [33,34]. The enrichment of CSCs at the border is apparent in the simulations both at large and small values of the diffusion coefficient *D*, although it is more marked in the former case. However, the reason for it seems to be slightly different in the two cases. At large *D*, the concentration of molecule is diluted out fast to the surrounding of the cellular system, displaying a radial profile that decreases moving away from the center and leaving the border with a concentration low enough to allow phenotypic switching. On the other hand, at low values of *D*, a transient initial enrichment propagates in time and in space along the radial direction, and is maintained at all later times. The diffusion constant *D* might play an important role also for possible therapeutic strategies based on mir222, since the size of carrier would determine the value of *D* and consequently the dynamics of phenotypic switching.

In conclusions, our paper elucidates the molecular mechanism of phenotypic plasticity in melanoma cells and provides guidance for therapeutic interventions targeting the switching into CSCs. To this end, it would be interesting to corroborate our results in vivo following the diffusion of fluorescent-tagged sh-mir222 and tracking the location of CXCR6 positive cells by immunohistochemistry.

## 4. Materials and Methods

### 4.1. The Computational Model

We construct a mean-field model in which CSCs are capable of unlimited proliferation [1,23] and can undergo different kinds of cellular division, into two CSCs, into two CCs, or into one CSC and one CC. On the other hand, CCs can only divide into two CCs for a limited number *g* of generations, after which they become senescent and die [23]. We also assume that CCs can switch stochastically to CSC at constant rate [35]. The new hypothesis of the current model with respect to previous models is that CSCs produce a molecular species that inhibits switching of CCs into CSCs according to a Michaelis–Menten molecular mechanism. We will discuss the experimental evidence for this in the second part of the paper.

We first introduce a delay in the action of CCs, representing the cellular replication and switch according to experimental data [1]. Under well-mixed microenvironment hypothesis, the model can be expressed by a system of delay rate equations in the concentration *S* of CSCs, the number $N_{j}$ of CCs at generation *j* (with 0≤j≤g) and the number *m* of the inhibiting species. The overall mechanism is sketched in Figure 1a and its dynamics can be described by the following system of equations
(3)$$\left\{\begin{array}{cc} \dot{m} & =k_{mp}S-k_{m}m\\ \dot{S} & =(k_{2S}-k_{2N}S+\sigma \sum_{i=0}^{g-1}N_{i}\left(t-\tau \right)\left(1-\frac{m^{h}\left(t-\tau \right)}{K^{h}+m^{h}\left(t-\tau \right)}\right)\\ {\dot{N}}_{0} & =(2k_{2N}+k_{1N1S}S-k_{d}N_{0}-\sigma \left(1-\frac{m^{h}\left(t-\tau \right)}{K^{h}+m^{h}\left(t-\tau \right)}\right)N_{0}\left(t-\tau \right)\\ \cdots \\ {\dot{N}}_{i} & =2k_{d}N_{i-1}-k_{d}N_{i}-\sigma \left(1-\frac{m^{h}\left(t-\tau \right)}{K^{h}+m^{h}\left(t-\tau \right)}\right)N_{i}\left(t-\tau \right)\\ {\dot{N}}_{i+1} & =2k_{d}N_{i}-k_{d}N_{i+1}-\sigma \left(1-\frac{m^{h}\left(t-\tau \right)}{K^{h}+m^{h}\left(t-\tau \right)}\right)N_{i+1}\left(t-\tau \right)\\ \cdots \\ {\dot{N}}_{g} & =2k_{d}N_{g-1}-2k_{dfin}N_{g} \end{array}\right.$$
with initial conditions $S\left(0\right)=S^{0}$, $m\left(0\right)=m^{0}$ and the $N_{j}\left(t\right)=N^{0}$ for t≤0 are set to the same initial values for sake of simplicity.

The rate equations are solved numerically using a custom made C code. The numerical method used to solve the system of differential equations is a variable-coefficient linear multistep Adams method in Nordsieck form (from the GNU Scientific Library), which employs a variable time-step.

Some of the numerical values that define Equation (Equation 3) are known, at least as order of magnitude. The division rate of melanoma cells add the variable, as in × 0.5–0.6/day. Typical initial populations of CCs is typically $N^{0}$∼10${}^{6}$ and that of CSCs is approximately one hundredth of it [1,23,36].

For the two-dimensional model we solve a discretized version of the diffusion equations
(4)$$\frac{\partial m(r,t)}{\partial t}\approx \frac{\Delta m(r,t)}{\Delta t}=k_{np}S\left(r,t\right)-k_{m}m\left(r,t\right)+\nabla^{2}m\left(r,t\right),$$
where m(r,t) is the number of molecules at position *r* of the lattice at time *t* and Δt = 3 × 10${}^{-4}$ days is the time step used for numerical differentiation, following a five-point stencil finite-difference method.

### 4.2. Cell Culture

Human IGR39 melanoma cells were obtained from Deutsche Sammlung von Mikroorganismen und Zellkulturen GmbH Cells were cultured in standard condition in DMEM (Dulbecco’s Modified Eagle’s Medium-high glucose, Cat. No ECB7501L, EuroClone, Italy) supplemented with 10% fetal bovine serum (FBS, Cat. No ECS0180D, EuroClone, Italy), 1% non-essential amminoacids (NEA, Cat. No ECB3054D, EuroClone, Italy), 1% antibiotics (Penicillin/Streptomycin, Pen/Strep, Cat. No ECB3001D, EuroClone, Italy), 1% L-glutamine (L-Glut, Cat. No ECB3000D-20, EuroClone, Italy) (complete medium) at $37{\;}^{\circ }$C and 5% CO${}_{2}$ and 95% humidity.

### 4.3. Conditioned Media

Cells were growth to 90% confluence and maintained for 16 h in the standard growth medium condition without 10% FBS. The media were collected and after a brief centrifugation to discard dead cell and debris, the supernatant was stored at −80${\;}^{\circ }$C for no more than 1 month until use.

### 4.4. FACS Analysis

Cells were analyzed for phycoerythrin (PE) anti-human CXCR6 (cod. MAB699, R&D System, Minneapolis, MN, USA). Non-specific mouse IgG is used as isotype control (Isotype). Non-specific mouse IgG2B (R&D, Minneapolis, cod.IC004IP) is used as isotype control (Isotype). For each flow cytometry evaluation, a minimum of $10^{6}$ cells were stained and at least 50,000 events were collected and analyzed. Flow cytometry and analysis was performed using a Gallios flow cytometer (Beckman Coulter Indianapolis, IN, USA).

### 4.5. Sponge Lentiviral Plasmids and Plasmid Amplification, Virus Production and Infection, Isolation and Detection of miRNAs

Sponge lentiviral plasmids were constructed with 8 inhibitors sequences each for hsa-miR-222-5p (5’-AGGATCTACAACCCTACTGAG-3’, pEGFP–Sponge-mir222-5p) by Creative Biogene (cod. PSE2774, Creative Biogene, Shirley, NY, USA). An miRNA control sponge plasmid, with 8 inhibitor sequences, was constructed for the scramble control sequence (5’-TTCTCCGAACGTGTCACGTGCT-3’, pEGFP–scramble) (cod. PSE2775, Creative Biogene, Shirley, NY, USA). The identity of 8 inhibitors sequences was verified by sequencing (Eurofins Genomics Service, Italy) using the primers reported in Table A2. EGFP was used as reporter gene, with puromycin as resistance marker and ampicillin resistance for bacterial amplification. E. coli One Shot TOP10 bacteria (cod. C404006, Invitrogen, Thermo Fisher Scientific, Waltham, MA, USA) were used for transformation with pEGFP–Sponge-mir222-5p or pEGFP–scramble lentiviral plasmid. Competent cells were expanded and selected in Luria Broth medium (cod. 12795-027, Invitrogen, Thermo Fisher Scientific, Waltham, MA, USA), containing 1 μg/mL Ampicillin (cod. A5354, Sigma Aldrich, Merck, Darmstadt, Germany) for 18 h at $37{\;}^{\circ }$C. Plasmid DNA extraction was performed by PureYield Plasmid Miniprep System (cod. A1223, Promega, Madison, WI, USA) according to the manufacturer’s instructions. pEGFP-Sponge-mir222-5p or pEGFP–scramble virus were produced by cotransfection of HEK293T cells with pCMV-dR8.2 (cod. 8455, Addgene, Watertown, MA, USA) and pCMV-VSV-G (cod. 8454, Addgene, Watertown, MA, USA). Transfections were carried out using calcium phosphate transfection method [37]. Virus was harvested 72 h post-transfection and infections were carried out in the presence of 10 μg/mL of polybrene. Cells were spinned at 2000 RPM for 1 h at RT. Following transduction, cells were selected with 1 μg/mL puromycin.

### 4.6. miRNA Analysis by miRCURY LNA Universal RT miRNA PCR

Total RNA was extracted using the standard RNA extraction method with TRIzol${}^{TM}$ (Invitrogen, Thermo Fisher Scientific, Waltham, MA, USA). RNA concentration in each sample was assayed with a ND-1000 spectrophotometer (NanoDrop, Thermo Fisher Scientific, Waltham, MA, USA). Reverse transcription reactions were performed using the miRCURY LNA RT Kit (cod. 339340, Exiqon, Qiagen, Germany). Reverse transcription thermocycling parameters were as follows: 42${\;}^{\circ }$C for 60 min and $95{\;}^{\circ }$C for 5 min. Real time q-RT-PCR analysis was performed using ViiA 7 Real-Time PCR System (Applied Biosystems, Thermo Fisher Scientific, Waltham, MA, USA). The PCR reaction included 1 ng of template cDNA, 1 μL of LNA PCR primers mix, 1 μL of RNAse-free water and 5 μL of miRCURY SYBR Green Master Mix (cod. 339345, Exiqon, Qiagen, Germany) in a total volume of 10 μL. Cycling conditions were as follows: 95${\;}^{\circ }$C enzyme activation for 10 min, followed by 50 cycles of amplification: 10${}^{\prime \prime }$ at 95${\;}^{\circ }$C for denaturing, 1 min at 60${\;}^{\circ }$C for annealing/elongation. Melting curve analysis was performed between 60${\;}^{\circ }$C and 95${\;}^{\circ }$C at a ramp rate of 0.11${\;}^{\circ }$C/s.

### 4.7. Isolation and Detection of miRNAs

Total RNA was extracted using the standard RNA extraction method with TRIzol${}^{TM}$ (Invitrogen, Thermo Fisher Scientific, Waltham, MA, USA). RNA concentration in each samples was assayed with a ND-1000 spectrophotometer (NanoDrop, Thermo Fisher Scientific, Waltham, MA, USA) and its quality assessed with an Agilent 4200 TapeStation or Bioanalyzer 2100 (Agilent Technologies, Santa Clara, CA, USA). Next generation sequencing experiments on infected cell lines was performed by Genomix4Life (Baronissi, Salerno, Italy). Indexed libraries were prepared from 1 μg/ea purified RNA with TruSeq SmallRNA Library Prep Kit (RS-200-0012, Illumina, San Diego, CA, USA) according to the manufacturer’s instructions. Libraries were quantified using the Agilent 4200 TapeStation (Agilent Technologies, Santa Clara, CA, USA) and pooled such that each index-tagged sample was present in equimolar amounts, with a final concentration of 2 nM for pooled samples. The pooled samples were subject to cluster generation and sequencing using an Illumina NextSeq 500 System (Illumina, San Diego, CA, USA) in a 1 × 75 single end format. The raw sequence files generated underwent quality control analysis using FastQC [38].

### 4.8. miRNA Analysis

Sequencing Illumina Indexes were trimmed using Trimgalore (v. 0.4.4) and FastQC (v0.11.7) (http://www.bioinformatics.babraham.ac.uk/publications.html). Trimmed sequences that aligned to human rRNAs using Bowtie (v. 1.2.2) ([39]) were discarded. Unmapped sequences were aligned on Human Genome (GRCh38, GCA_000001405.15, https://www.ncbi.nlm.nih.gov/) and miRBase library (v.22 [40]) using miRDeep2 (v. 0.1.1 [41]). Estimated counts were averaged for miRNAs mapping on multiple hairpins and normalized over total number of counts.

### 4.9. miRNA Differential Expression Analysis

Relative expression values of miRNAs are computed taking the average of the scramble as reference values. We filtered the initial pool of 2656 miRNAs as follows: First, we discard miRNAs with absolute expression below $10^{3}$ in any of the samples. Then, we keep only those miRNAs whose expression displays changes of 10% with respect to the Scramble samples. Finally, we keep only those miRNAs that display a change of at least 25% with respect to Scramble in at least one of the Sponge samples.

### 4.10. miRNAs Expression in Conditioned Media

MiRNAs were extracted from the conditioned media collected as described in the materials and method section by miRCURY RNA Isolation kit-Biofluids (cod. 300112, Exiqon, Qiagen, Germany). According to the manifacturer’s instructions, the samples were treated with DNase and eluted with 20 μL nuclease-free water. The quality of RNA extracted was carried out by capillary electrophoresis using Bioanalyzer (RNA 6000 Pico Kit, Agilent Technologies, Santa Clara, CA, USA). The libraries were obtained using the kit TruSeq Small RNA Library Prep (RS-200-0012, Illumina, San Diego, CA, USA) and validated by Bioanalyzer using Agilent High Sensitivity DNA kit (cod. 5067-4626, Agilent Technologies, Santa Clara, CA, USA). A pick of about 140 bp was expected corresponding to miRNA, since during the preparation of the libraries we added two 60 bp adapters to each end of the existing 22 bp miRNAs sequence. To enrich the libraries in the fraction of the miRNAs only, we carried out an 6% polyacrilamide electrophoresis (gel Novex^TM^ TBE, Invitrogen, Thermo Fisher Scientific, Waltham, MA, USA) and we cut the band between the two markers of 145 and 160 bp. We then validated the libraries and quantified by Bioanalyzer using Agilent High sensitive DNA (cod. 5067-4626, Agilent Technologies, Santa Clara, CA, USA). The library was then sequenced by Genomix4Life (Baronissi, Salerno, Italy) using Illumina NextSeq 500 System (Illumina, San Diego, CA, USA).

### 4.11. Migration Assay

A wound was introduced in the central area of the confluent cell sheet by using a pipette tip and the cell migration was followed in time-lapse using a Leica DM IL LED inverted microscope (Leica, Germany) at $37{\;}^{\circ }$C and 5% CO${}_{2}$ and 95% humidity (UNO stage top incubator, Okolab, Italy) every 15 min until the complete closure of the scratch with a 10× objective.

### 4.12. PIV Analysis

Velocity field was estimated using PIVlab app for Matlab [42]. The method is based on the comparison of the intensity fields of two consequent images. The difference in the intensity is converted into velocity field measured in px/frames and then converted to μm/h [43].

### 4.13. Spheroids Culture, Collagen Invasion Assay and Image Analysis

Multicellular spheroids were obtained from sub-confluent cells using the hanging-drop technique (2% methylcellulose, cod. M6385, Sigma Aldrich, Merck, Darmstadt, Germany; 500 cells/50 μL drop; 7 days spheroid assembly) according to Ref. [44]. Spheroids were embedded in non-pepsinzed rat-tail collagen type I solution (2.5 mg/mL; BD Biosciences, Franklin Lakes, NJ, USA) prior to collagen polymerization (37 ${}^{\circ }$C for 5–10 min). The collagen matrix invasion process was monitored by bright-field time-lapse microscopy using a LEICA DMI8 microscope (Leica, Germany) acquiring one frame per hour for 72 h. The image analysis is done using Matlab Image Processing Toolbox. At first, the image is cut to focus on the area of interest. Then it is thresholded and a black-white image is created using built-in Matlab function “edge”. After it is dilated with “imdilate” and the holes are filled with “imfill”. Small noise is removed by “bwareaopen”. Then the image is eroded and filled by “imerode” and “imfill” functions. In the resulting images the center of the white (detected) area is found. Then, we compute the average distance from the center to all other detected points. This value corresponds to the spread at a given time step.

### 4.14. Statistical Analysis

For miRNA expression, *p*-values were obtained from EdgeR differential analysis [45]. For all the other cases, the two-tail unpaired *t*-test was used to assess statistical significance.

## Figures and Tables

**Figure 1 jcm-09-02573-f001:**
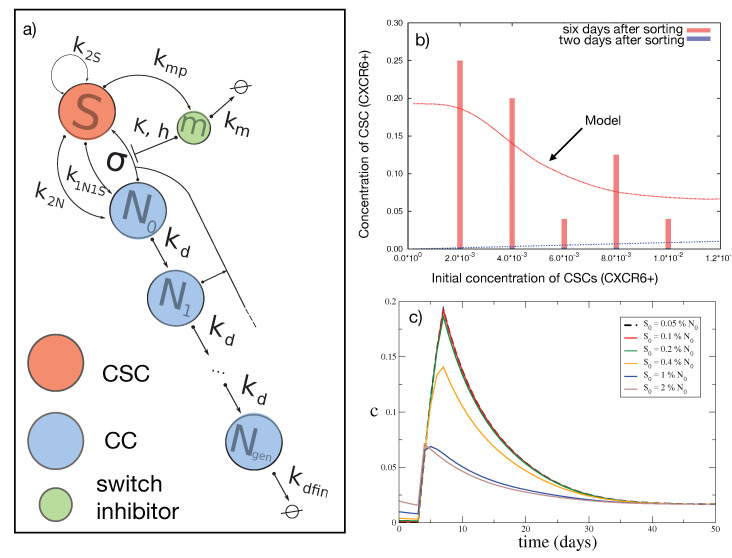
Mean-field model (**a**) Reaction network for the mean-field model, involving cancer stem cells (CSCs) (*S*), cancer cells (CCs) ($N_{j}$) and inhibitor molecule *m*. (**b**) The experimental fraction c(t) of CSCs after 2 days (blue bars) and after 6 days (red bars) from their sort out, as a function of their remaining fraction c(0). The curves indicate the simulated results. (**c**) The dynamics of the concentration c(t) of CSCs with different choices of the initial value c(0) ($N_{0}=10^{6}$, $m_{0}=0$).

**Figure 2 jcm-09-02573-f002:**
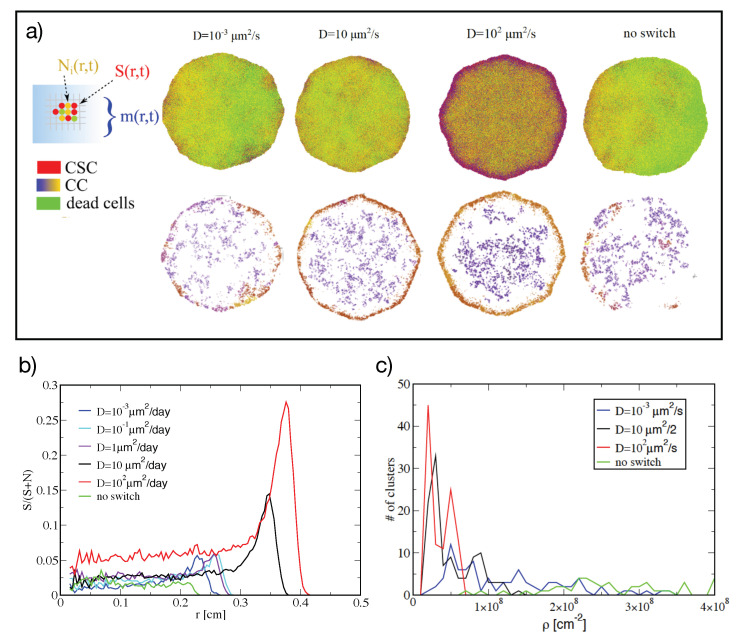
Two-dimensional model (**a**) A sketch of the lattice model used for the calculations is shown on the left. Top panels show snapshots displaying the spatial arrangements of CCs of different ages and of CSCs for different values of the diffusion constant *D*. Below each panel, we display CSCs clusters, subtracted their density halo and colored it differently for each cluster for the same values of *D*. All the panels are obtained after 30 days of growth. (**b**) The radial density profile of the fraction of CSCs after 30 days of growth, with τ=4, for different values of *D* and for the model without switching. (**c**) The histogram of densities of the different clusters.

**Figure 3 jcm-09-02573-f003:**
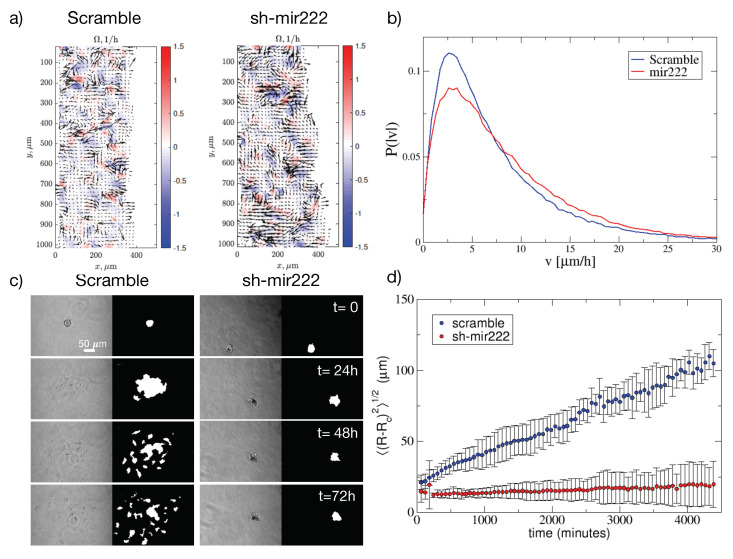
Impact of mir222-5p knock-down on cell migration. (**a**) PIV analysis of wound healing assays for IgR39 cells transfected with hsa-miR-222-5p inhibitor (sponge miR222 [sh-mir222]) or scramble sequence (scramble) as control. Arrows indicate the local velocity and the color represents the local vorticity calculated with PIV as discussed in the Materials and Methods section. (**b**) The distribution of local velocities shows a reduction of the velocities upon mir222-5p knock down. (**c**) Growth of 3D spheroids in a collagen network sh-mir222 and scramble IgR39 cells. After digitization, we quantify the area occupied by cells as described in the Materials and Methods section. (**d**) The spread of the spheroid is quantified by measuring ${\left(\langle {\left(R-R_{c}\right)}^{2}\rangle \right)}^{1/2}$, where $R_{c}$ is the center of mass of the spheroid. Spreading of sh-mir222 cells is strongly impaired.

**Figure 4 jcm-09-02573-f004:**
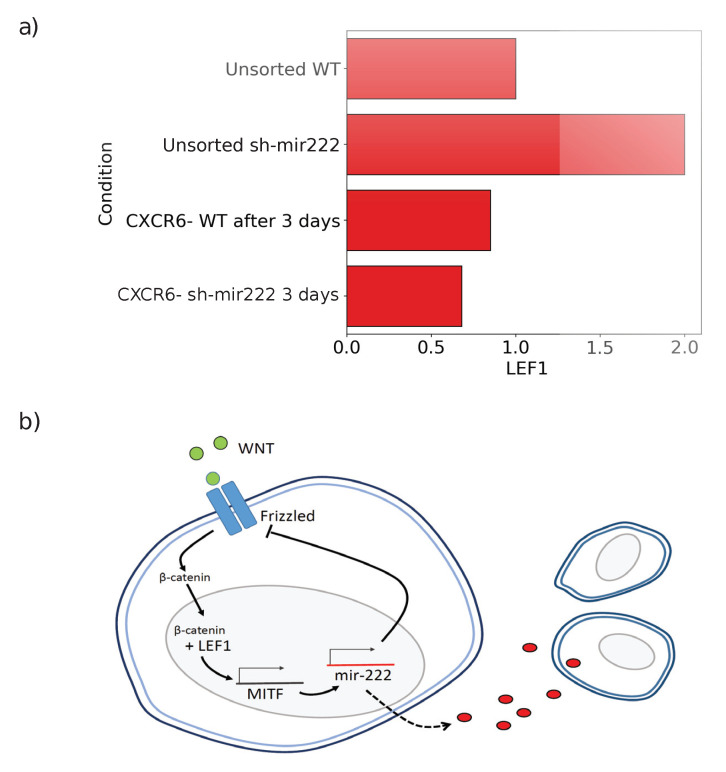
Impact of mir222-5p on the Wnt pathway. (**a**) The level of expression of LEF1 in different conditions in IgR39 cells, Results from [1]. The reported values are normalized with respect to the value of LEF1 in unsorted WT condition, whose value is therefore set to one. (**b**) A cartoon highlighting the role of mir222 on the Wnt pathway. Binding of Wnt to Frizzled leads to the stabilization of β-catenin and its translocation to the nucleus. Here, β-catenin can interact with the transcription factor LEF1, activating it. LEF1 allows the expression transcription factor MITF and, as a consequence, of MITF targets, including mir222. The latter interact with Frizzled, repressing Fzd-7 mRNA translation. Mir-222 is also released outside the cell (dashed black line) acting on paracrine cells.

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
