# Peer review of "MicroRNA-222 Regulates Melanoma Plasticity"

_jcm, 2020, doi:10.3390/jcm9082573_

Round 1
Reviewer 1 Report
Lionetti and colleagues study the role of miR-222 in the possible switching between cancer cells to cancer stem cells, and how melanoma cells maybe treated.
This is a very well written manuscript, my only comment will be to add all equations into methods sections and remove from the results, this will make current more accessible to a general audience.
Author Response
REFEREE 1
This is a very well written manuscript, my only comment will be to add all equations into methods sections and remove from the results, this will make current more accessible to a general audience.
We are grateful for his/her valuable comments. We moved the description of the model in the Materials and Methods section as suggested.
Reviewer 2 Report
The study of Lionetti et al., focuses on mathematical modeling of the phenotypic switching occurring in the melanoma cells. Report presents preliminary in vitro data as the background to future experiments. Melanoma is highly heterogenous tumor and melanoma plasticity is one of the factors responsible for resistance to treatment, that makes the study of Lionetti et al interesting. However, I have several concerns about the clarity of the manuscript and the authors should properly address the following specific comments.
- “Discussion” paragraph needs new organization I would also recommend a more rigorous discussion of literature data, especially concerning WNT signaling pathway. The mechanism of activation of WNT signaling is much more complex than presented in the manuscript, especially in Figure 4. Activation of canonical WNT signaling occurs upon binding of WNT ligand to Frizzled and LRP5/6 receptors, then beta-catenin is stabilized and is transferred from the cytoplasm to the nucleus where it can interact with several proteins, e.g. LEF1. Then beta-catenin serves as a transcriptional regulator of the expression of WNT target genes e.g. MITF, so MITF expression is mediated by the complex of two key effectors of the WNT signaling pathway, LEF1 and β-catenin. In my opinion, authors simplify the mechanism of phenotypic switch of melanoma cells.
- Explain what does (cf. Fig. 2a) line 143 mean
- Put a missing dot after Fig 1a
4.Instead of “In Ref. [1],” in line 224 better formulation should be used e.g. “we reported (…) [1].
5. “A A cartoon highlighting the role of mir222 on the Wnt pathway.”-authors should remove one “A”
Moreover, the manuscript would certainly benefit from extensive editing by an English native speaker, as there are stylistic mistakes, making the article difficult to read. Therefore, I recommend the authors to look for a relevant help.
Author Response
REFEREE2
The study of Lionetti et al., focuses on mathematical modeling of the phenotypic switching occurring in the melanoma cells. Report presents preliminary in vitro data as the background to future experiments. Melanoma is highly heterogenous tumor and melanoma plasticity is one of the factors responsible for resistance to treatment, that makes the study of Lionetti et al interesting.
However, I have several concerns about the clarity of the manuscript and the authors should properly address the following specific comments.
We thank the referee for the positive comments, we have carefully considered her/his suggestions, and made revisions to our work accordingly.
- “Discussion” paragraph needs new organization I would also recommend a more rigorous discussion of literature data, especially concerning WNT signaling pathway. The mechanism of activation of WNT signaling is much more complex than presented in the manuscript, especially in Figure 4. Activation of canonical WNT signaling occurs upon binding of WNT ligand to Frizzled and LRP5/6 receptors, then beta-catenin is stabilized and is transferred from the cytoplasm to the nucleus where it can interact with several proteins, e.g. LEF1. Then beta-catenin serves as a transcriptional regulator of the expression of WNT target genes e.g. MITF, so MITF expression is mediated by the complex of two key effectors of the WNT signaling pathway, LEF1 and β-catenin. In my opinion, authors simplify the mechanism of phenotypic switch of melanoma cells.
We agree with the referee that the WNT pathway is a highly complex pathway, involving a large number of players. Figure 4 was intended to be a schematic representation of those elements in WNT pathway that to are necessary to illustrate the feedback-loop mechanism. The changes in beta-catenin expression levels and other WNT-related pathway were reported in a previous work of our group (Sellerio et al. Sci. Rep. 2015, ref 1 in the main text). We have now included in the discussion a more detailed description of the WNT pathway, including the role of beta-catenin in the feedback-loop and revised in Figure 4b in order to reflect this point.
Explain what does (cf. Fig. 2a) line 143 mean
We clarify better in the text the reference to Fig.2a
- Put a missing dot after Fig 1a
Done.
4.Instead of “In Ref. [1],” in line 224 better formulation should be used e.g. “we reported (…) [1].
Done
- “A A cartoon highlighting the role of mir222 on the Wnt pathway.”-authors should remove one “A”
Done.
Moreover, the manuscript would certainly benefit from extensive editing by an English native speaker, as there are stylistic mistakes, making the article difficult to read. Therefore, I recommend the authors to look for a relevant help.
We have now extensively revised the text as suggested by the referee.
Reviewer 3 Report
Herein, the authors describe how hsa-mir-222-5p plays a critical role in Wnt signalling of human melanoma cells (IgR39 cells) which is important in regulating the epithelial-mesenchymal transition. The authors describe a new model in which all IgR39 cancer cells can spontaneously switch to cancer stem cells and that this pathway is inhibited by specific molecular species produced by CSC’s. They demonstrated this by sorting out CSC’s from IgR39 cells using anti-human CXCR6-PE and observed that the remaining CC’s spontaneously and massively switch to CSC’s. The authors also investigated the role hsa-mir222-5p as a key factor in IgR39 cells switching from CC’s to CSC’s. By knocking down hsa-mir222-5p using the miRNA sh-mir222, they observed that the cells lost the ability to express CSC markers and acquired a less aggressive phenotype in a 2D wound healing assay and in 3D spheroids. The authors also used mathematical modelling to predict how different diffusion rates of sh-mir-222, presumably relating to the method in which the miRNA is delivered, affect CSC switching and their predicted location within the tumor niche.
Overall I really enjoyed reading this paper, the assays are well done and the mathematical models were interesting to learn about. The conclusions of this paper are novel and very relevant to what most cancer researchers and clinicians are investigating in the hunt to find novels therapies that can synergize with the successful (yet limited in some ways) immunotherapies available. Most of the manuscript was very well written however some sections (abstract) require grammatical corrections and spell check.
It’s not clear from the text how CSCs are phenotypically identified (I can see it in the mat and methods and in some figures), but it should be stated somewhere in the text why this marker was chosen and include a reference to support the use of this marker to sort the CSCs from the CC’s.
Line 1: change ‘tumor’ to ‘tumors’
Line 2: suggest rewording to “Cell plasticity in melanoma is one of the main culprits behind its metastatic capabilities.”
Line 4: change to “[…] melanoma cells line […]”
Line 5: change ‘show’ to ‘shows’
Line 126: insert ‘sites’: “[…] all the neighboring sites are occupied […]”
Line 131: Suggest changing “coated into a vesicle” to “encapsulated into a vesicle”
Line 157: change ‘ad’ to ‘at’
Line 167: add ‘d’ to conditioned
Line 168: change to “[…] we knocked it down in […]”
Line 171: Only one CSC marker is included in this manuscript, suggest rewording to “a phenotypic marker of CSCs, CXCR6, was expressed […]”
Line 180: delete ‘this’
Line 183: add a comma “[…] knockdown, velocities […]”
Figure 3 caption. Suggest rewording to indicate how the cells were treated, ie.:
“PIV analysis of wound healing assays for IgR39 cells transfected with sh-mir222 or the control, scramble.”
Be sure that each reagent and piece of equipment has both a company name and a country, missing in many parts throughout the Mat and Methods.
Make uniform whether to place a space between # and it’s unit or not, it varies throughout.
Ie 1μl or 1 μl in lines 294.
Line 263: swap words: “to discard cell debris and dead cells, and the […]”
Line 273: Change to “A miRNA control sponge plasmid, with 8 inhibitor sequences, was constructed for the scramble […]”
Line 275: Change to “[…] was verified by sequencing […]”
Line 289: Add TM and correct capitalization “TRIzolTM “
Line 290: Add company and country for the spectrophotometer
Line 300: Add superscript TM to TRIzolTM.
Line 307: Change to “[…] with a final concentration of 2nM for pooled samples.”
Line 308: sampels to samples.
Line 309: sequance to sequence.
Line 317: harpins to hairpins
Line 333: miRNAS to miRNAs, also consider rewording “we added two 60bp adapters to each end of the existing 22bp mRNA sequence.”
Line 340-41: “A wound was introduced in the central area of the confluent cell sheet by using a pipette tip, and the cellular migration was followed by […]”
Line 345: change were to was
Line 359: remove the ‘s’ from values
Figure A4: “[…] according to the Materials and Methods section […]” and the last sentence needs to be reworded, possible option “Cells sponged with a scramble miRNA was used as a control.”
Figure A5: Flow cytometry data is better presented as dot plots where we can visualize the population of cells and see the relative MFI, was there a difference in MFI after treatment?
I agree with the concluding sentence that the paper gives evidence for the use of knocking down has-mir-222 in melanoma therapy, but I would like to see an elaboration of the “guidance for therapeutic interventions”. I agree that the authors devised a clever mathematical model that can predict where CSC clustering would occur dependent on high or low D. I would like further explanation in the discussion section highlighting how various in vivo delivery methods would relate to high or low D. There was a brief mention earlier on about vesicles, this could be discussed as could other options including the effect of changing the size of the molecule (add a tag) or using a melanoma specific antibody to target the tumor.
I would also suggest perhaps a brief mention of future studies as I’m intrigued by what studies would come next, much more could be done in an animal model at this point. Ideas that come to mind include using a fluorescent tagged-sh-mir222 and looking at its diffusion rate using in vivo imaging in a melanoma tumor bearing mouse model with treatments at various time-points after tumor injection. Also could use immunohistochemistry to track the expression of CXCR6 and rate of CSC switching as well as see their location. Lastly a discussion of how sh-mir-222 could be used in therapy, any considerations for its use, such as the route of delivery (outside or inside a tumor), number of doses (how fast is the molecule is cleared from the body) etc.
Author Response
REFEREE3
Herein, the authors describe how hsa-mir-222-5p plays a critical role in Wnt signalling of human melanoma cells (IgR39 cells) which is important in regulating the epithelial-mesenchymal transition. The authors describe a new model in which all IgR39 cancer cells can spontaneously switch to cancer stem cells and that this pathway is inhibited by specific molecular species produced by CSC’s. They demonstrated this by sorting out CSC’s from IgR39 cells using anti-human CXCR6-PE and observed that the remaining CC’s spontaneously and massively switch to CSC’s. The authors also investigated the role hsa-mir222-5p as a key factor in IgR39 cells switching from CC’s to CSC’s. By knocking down hsa-mir222-5p using the miRNA sh-mir222, they observed that the cells lost the ability to express CSC markers and acquired a less aggressive phenotype in a 2D wound healing assay and in 3D spheroids. The authors also used mathematical modelling to predict how different diffusion rates of sh-mir-222, presumably relating to the method in which the miRNA is delivered, affect CSC switching and their predicted location within the tumor niche.
Overall I really enjoyed reading this paper, the assays are well done and the mathematical models were interesting to learn about. The conclusions of this paper are novel and very relevant to what most cancer researchers and clinicians are investigating in the hunt to find novels therapies that can synergize with the successful (yet limited in some ways) immunotherapies available. Most of the manuscript was very well written however some sections (abstract) require grammatical corrections and spell check.
We are grateful to the reviewer for the positive remark on our manuscript, we have revised the manuscript as suggested.
It’s not clear from the text how CSCs are phenotypically identified (I can see it in the mat and methods and in some figures), but it should be stated somewhere in the text why this marker was chosen and include a reference to support the use of this marker to sort the CSCs from the CC’s.
Identification of CSC was based on the CXCR6 marker that was previously characterized and validated by our group. We now report explicitly this information in the introduction.
Line 1: change ‘tumor’ to ‘tumors’
Done.
Line 2: suggest rewording to “Cell plasticity in melanoma is one of the main culprits behind its metastatic capabilities.”
Done
Line 4: change to “[…] melanoma cells line […]”
Done
Line 5: change ‘show’ to ‘shows’
Done
Line 126: insert ‘sites’: “[…] all the neighboring sites are occupied […]”
Done
Line 131: Suggest changing “coated into a vesicle” to “encapsulated into a vesicle”
Done
Line 157: change ‘ad’ to ‘at’
Done
Line 167: add ‘d’ to conditioned
Done
Line 168: change to “[…] we knocked it down in […]”
Done
Line 171: Only one CSC marker is included in this manuscript, suggest rewording to “a phenotypic marker of CSCs, CXCR6, was expressed […]
Done
Line 180: delete ‘this’
Done
Line 183: add a comma “[…] knockdown, velocities […]”
Done
Figure 3 caption. Suggest rewording to indicate how the cells were treated, ie.:
Done
“PIV analysis of wound healing assays for IgR39 cells transfected with sh-mir222 or the control, scramble.”
Done
Be sure that each reagent and piece of equipment has both a company name and a country, missing in many parts throughout the Mat and Methods.
Done
Make uniform whether to place a space between # and it’s unit or not, it varies throughout. Ie 1μl or 1 μl in lines 294.
Done
Line 263: swap words: “to discard cell debris and dead cells, and the […]”
Done
Line 273: Change to “A miRNA control sponge plasmid, with 8 inhibitor sequences, was constructed for the scramble […]”
Done
Line 275: Change to “[…] was verified by sequencing […]”
Done
Line 289: Add TM and correct capitalization “TRIzolTM “
Done
Line 290: Add company and country for the spectrophotometer
Done
Line 300: Add superscript TM to TRIzolTM.
Done
Line 307: Change to “[…] with a final concentration of 2nM for pooled samples.”
Done
Line 308: sampels to samples.
Done
Line 309: sequance to sequence.
Done
Line 317: harpins to hairpins
Done
Line 333: miRNAS to miRNAs, also consider rewording “we added two 60bp adapters to each end of the existing 22bp mRNA sequence.”
Done
Line 340-41: “A wound was introduced in the central area of the confluent cell sheet by using a pipette tip, and the cellular migration was followed by […]”
Done
Line 345: change were to was
Done
Line 359: remove the ‘s’ from values
Done
Figure A4: “[…] according to the Materials and Methods section […]” and the last sentence needs to be reworded, possible option “Cells sponged with a scramble miRNA was used as a control.”
We thank the reviewer for carefully reading the manuscript. We modified and reviewed the text according to reviewer suggestions.
Figure A5: Flow cytometry data is better presented as dot plots where we can visualize the population of cells and see the relative MFI, was there a difference in MFI after treatment?
Exemplificative dot plots from our FACS experiments are now reported in supplementary Figure 5, along with evaluated median fluorescence for each condition. The decrease of fluorescence in sh-mir222 cells stained for CXCR6 compared to unstained cells confirm the absence of the marker in these cells.
I agree with the concluding sentence that the paper gives evidence for the use of knocking down has-mir-222 in melanoma therapy, but I would like to see an elaboration of the “guidance for therapeutic interventions”. I agree that the authors devised a clever mathematical model that can predict where CSC clustering would occur dependent on high or low D. I would like further explanation in the discussion section highlighting how various in vivo delivery methods would relate to high or low D. There was a brief mention earlier on about vesicles, this could be discussed as could other options including the effect of changing the size of the molecule (add a tag) or using a melanoma specific antibody to target the tumor.
We briefly discuss this point.
I would also suggest perhaps a brief mention of future studies as I’m intrigued by what studies would come next, much more could be done in an animal model at this point. Ideas that come to mind include using a fluorescent tagged-sh-mir222 and looking at its diffusion rate using in vivo imaging in a melanoma tumor bearing mouse model with treatments at various time-points after tumor injection. Also could use immunohistochemistry to track the expression of CXCR6 and rate of CSC switching as well as see their location. Lastly a discussion of how sh-mir-222 could be used in therapy, any considerations for its use, such as the route of delivery (outside or inside a tumor), number of doses (how fast is the molecule is cleared from the body) etc
We discuss briefly possible future studies in vivo, as suggested by the Referee. We find, however, a detailed discussion of therapeutic strategies to be premature at this stage.